# Numerical Study on the Heating Effect of a Spring-Loaded Actuator—Part II: Optimization Design of Heater Parameters

Zhen Zhao, Lei Xi *, Jianmin Gao, Liang Xu and Yunlong Li

State Key Laboratory for Manufacturing Systems Engineering, Xi'an Jiaotong University, Xi'an 710049, China; zhaozhen.900803@stu.xjtu.edu.cn (Z.Z.)

* Correspondence: xilei100@mail.xjtu.edu.cn

**Abstract:** Unfavorable temperatures and humidity will cause the failure of spring actuators. In order to ensure the safe operation of the actuator, it is necessary to optimize the design of the built-in heater system of the actuator itself. In this study, an experimental design and a response surface model were used to fit the empirical formulas for the minimum temperature, maximum humidity, and maximum temperature on the heater surface. On this basis, a genetic algorithm was used to establish the optimal size of the heater in the chamber of the spring actuator. The study results show that the air inside the actuator shows a trend of a decrease in temperature and an increase in relative humidity from top to bottom. The empirical equation obtained by fitting the second-order response surface model has high accuracy, and the maximum prediction errors for the minimum temperature, maximum relative humidity, and maximum temperature of the heater surface of the spring actuator are −0.5%, 11.7%, and 4.7%, respectively. When the environmental temperature reduces from 313 K to 233 K, the optimal heating power of the heater increases from 10 W to 490 W, the optimal relative length increases from 3.57 to 6, and the optimal relative width increases from 1 to 5.3. Therefore, the study can act as a reference for the temperature and humidity control system of future actuators.

**Keywords:** spring actuator; temperature and relative humidity; heater; numerical simulation; optimization design

## 1. Introduction

The power system provides easy-to-use, efficient, and less polluting electrical energy that is widely used, promoting change in all areas of social production, and its size and technology have become one of the recognized signs of a country's level of economic development. High-voltage circuit breakers have the role of protecting and controlling the power system; their reliability is the basis for ensuring the safe and stable operation of the power system [1,2]. Previous studies have shown that many of the failures of high-voltage circuit breakers are caused by the failure of the operating mechanism [3,4]. Compared with other types of actuators, the spring actuator has the advantage of requiring a small capacity of power supply, and it can be used in both remote electric energy storage and manual operation; therefore, it is widely used in high-voltage circuit breakers [5]. Moreover, a typical spring actuator works in different positions and seasons when the environmental temperature and relative humidity are different. Therefore, there is an urgent need to study the laws of temperature and relative humidity and their effects on the operating mechanism and its internal parts to ensure its safe operation.

Many experts and scholars have explored the effects of temperature on the performance of actuators. In order to obtain the characteristics of shape-memory alloy spring actuators, Jianzuo et al. [6] investigated the output force and displacement of shape-memory alloy springs at different temperatures. The results show that the output displacement of the shape memory alloy spring actuator increases with an increase in temperature. Hyo et al. [7] evaluated the characteristics of shape memory alloy spring actuators and

performed isothermal loading and unloading, shape recovery force measurements, temperature tracking, and load tracking tests on the actuators. Hu et al. [8] proposed a position controller that includes anti-saturation and anti-overheating functions to control the position of the shape memory alloy actuator. The position control simulation model of the spring actuator was established and simulated. The results showed that the established model and control method could effectively control the position of shape memory alloy actuators. Degeratu et al. [9] conducted thermal analysis experiments on the spring actuator, in order to determine the transition temperature of the shape memory alloy spring actuator, and developed a comprehensive graphical interface based on the thermal analysis results. In order to bring the shape memory alloy spring mechanism to a given temperature, Park et al. [10] used hot and cold water to heat and cool the mechanism. The results show that when the temperature increases from 301 K to 355 K, the spring mechanism can generate a force of 130 N, which can achieve the flexion and extension motion of the arm at a 1-HZ driving speed. Degeratu et al. [11] performed a thermal analysis of the spring actuator material in order to improve the overall performance of the barrier structure, which was used to determine the operating time of this spring at different values of activation currents and phase transition rates. To develop more advanced sensor actuators, Holanda et al. [12] proposed the use of some smart materials that can change their mechanical properties when subjected to certain thermodynamic loads and explored the effects of unbalanced excitation forces and temperature control systems. Xiong et al. [13] established the thermodynamic theoretical model of an electrothermal shape memory alloy coil spring actuator under different conditions. They verified the thermodynamic characteristics of the actuator by numerical simulations and experimental tests and analyzed the temperature-force response and temperature-displacement response, as well as the force-displacement response at different temperatures.

Humidity can also have a strong effect on the actuator and the materials used for the actuator. Cabuz et al. [14] showed that humidity was the main reason for the failure of a touch-mode electrostatic actuator. Ryabchun et al. [15] explored the driving modes of an actuator material at different relative humidities and showed that the twisting, curling, and winding of the material varied with humidity. Arazoe et al. [16] and Xu [17] explored the performance changes of moisture-driven actuators when the relative humidity varied; the results showed that the actuators exhibited different performance values for different relative humidity variations. Wang et al. [18] fabricated a soft actuator with a two-layer structure under multiple stimulus responses, and the results showed that the actuator exhibits different actuation performance according to changes in temperature and humidity. It is also worth noting that this paper explored the optimization of the internal heater parameters of the spring mechanism at different temperatures and humidity, which is also helpful for future research in terms of applications in the fields of medicine [19], chemistry and biology [20,21], and micro- and nanofluidics [22–24].

From the above literature review, it can be seen that many experts and scholars have studied the effect of temperature on the actuator material, while studies on the control of temperature and relative humidity in the actuator chamber to ensure the safe operation of the actuator are rare. In this paper, a response surface model and a genetic algorithm are used to optimize the design of the heating system for controlling the temperature and relative humidity in the chamber of a typical spring actuator. Many experts and scholars have conducted studies on the prediction and optimization of actuators and in many other areas using genetic algorithms [25–27]. Dalla et al. [28] proposed a model-based fault detection and isolation method using genetic algorithms to identify fault precursors before the system's performance started to be compromised, in order to help detect initial faults in the flight control system, improve aircraft safety and reduce maintenance costs, and schedule maintenance interventions and actuator replacements in a timely manner. Lee et al. [29] used a genetic algorithm to optimize the design of a new electromagnetic engine valve, in order to improve the vibration frequency of the armature and reduce the transition time of the engine valve. The high performance of the new actuator was also

verified using dynamic finite element analysis. Foutsitzi et al. [30] used a genetic algorithm to optimize the design of the beam structure, in order to obtain the best voltage and the best placeholder for the beam structure. The optimal values of the best placement and voltage obtained by the optimization were applied to the piezoelectric actuator to minimize the error between the achieved and desired shapes.

In summary, any change in temperature and humidity will have a great impact on the performance of the actuator, and unreasonable temperature and humidity distribution is likely to cause a failure in the performance of the actuator; therefore, it is necessary to build a heater in the actuator to obtain a suitable temperature and humidity range, so as to ensure the safe operation of the actuator. In response to the above questions, for this paper, the temperature and relative humidity calculation and heater optimization design of a typical spring actuator heating system were carried out, based on numerical simulation. Firstly, the temperature and humidity distribution characteristics inside the spring actuator chamber were analyzed; then, the empirical correlation equations of the minimum temperature, $T_{min}$, maximum relative humidity, $RH_{max}$, and maximum heater surface temperature, $T_{heater}$, were obtained, based on the experimental design and response surface model fitting with respect to the environmental temperature $T$, heater power $Q$, heater relative length $L/H$, and heater relative width, $W/H$. The empirical correlation equations were then analyzed with an ANOVA for accuracy, and the significance of the research parameters was analyzed; finally, the heater design of the spring-operated mechanism was optimized, based on the empirical correlation equations and the genetic algorithm.

## 2. Research Subject

The research object of this paper is a typical spring actuator, the structure of which is shown in Figure 1, including the chamber, the springs, and the built-in heating system—the heater. The chamber is 1680 mm long, 784 mm wide, and 1225 mm high, and the parts inside the chamber need to work in an environment where the temperature is greater than 265 K, the relative humidity is less than 85%, and the maximum temperature of the heater surface is less than 340 K. Since the actuator faces a complex environment with large temperature and relative humidity differences, a heater needs to be fitted inside the chamber to ensure the normal operation of the spring actuator. The heater is arranged in the center of the right side of the chamber, 150 mm from the bottom, and the material is aluminum alloy. The height $H$ of the heater is constant at 55 mm, and the length $L$ and width $W$ of the heater are dimensionless, along with the relative length $L/H$ and the relative width, $W/H$. The environmental relative humidity in this study is constant at 95%, the environmental temperature $T$ varies from 233 K to 313 K, the heating power $Q$ varies from 10 W to 500 W, and the relative length $L/H$ of the heater varies from 2 to 6. The relative width $W/H$ varies from 1 to 5. The detailed research parameters are shown in Table 1. The response parameters include the minimum temperature, $T_{min}$, maximum relative humidity, $RH_{max}$, and the maximum heater surface temperature, $T_{heater}$, inside the actuator chamber.

**Table 1.** Research parameters.

| Variables | Minimum Value | Intermediate Value | Maximum Value |
|---|---|---|---|
| $T$(K) | 233 | 0 | 313 |
| $L/H$ | 2 | 4 | 6 |
| $W/H$ | 1 | 3 | 5 |
| $Q$(W) | 10 | 255 | 500 |

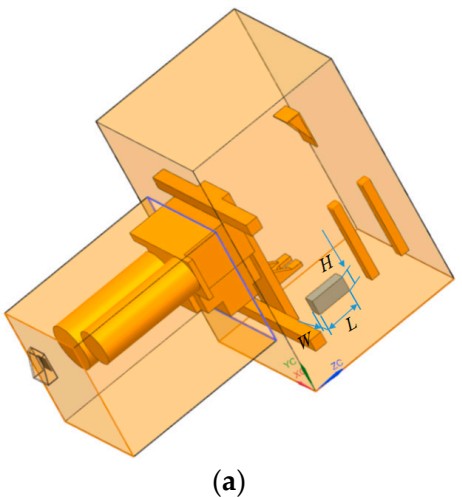
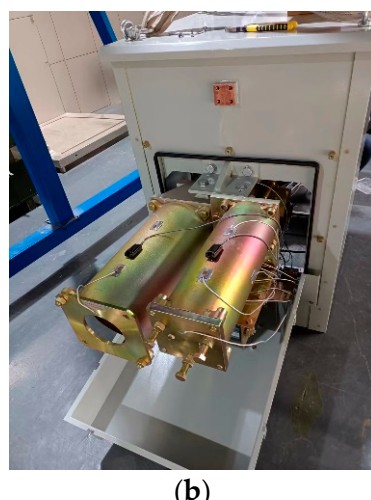

(**a**)　　　　　　　　　　　　　　　　　　　　　(**b**)

**Figure 1.** The research subject: (**a**) 3-D model; (**b**) physical model.

## 3. Research Methodology

### 3.1. Numerical Modeling Approach

In this paper, a fluid-solid coupling numerical simulation method was adopted to complete the calculation of the temperature and relative humidity distribution inside a typical spring actuator chamber. The numerical calculation model is shown in Figure 2, where Figure 2a,b shows the fluid-solid conjugate model and calculation flowchart, respectively. As can be seen from Figure 2b, the fluid-solid conjugate model includes the solid domain of the actuator shell, the components inside the chamber, the heater, and the air-fluid domain. The ANSYS fluent software (V18, ANSYS Inc., Pittsburgh, PA, USA) is used to solve the thermal conductivity equation of the solid domain. The driving force of the air-fluid domain is created by the change in density of the air after heating, whereby a natural convection flow is formed under the action of gravity [31]. Therefore, the fluid domain is assumed to be a three-dimensional, constant, naturally convective laminar flow with gravity. The finite volume method was utilized to calculate the mass, momentum, and energy conservation equation, and each term in the equations was in a high-precision discrete format. The three-dimensional compressible Reynolds time-averaged N–S equations were solved using Fluent. The intersection between the solid and fluid domains was set as a fluid-solid coupling surface, where both sides of this surface have the same temperature and heat flux. The convergence condition is reached when the residual level of each equation is below $10^{-6}$ and the numerical simulation is stopped. The continuity, momentum, and energy equations can be sourced from the literature [32,33]. According to the conservation equation for species transport, the amount of water vapor in the air can be calculated, after which the relative humidity value can be calculated from the air temperature and saturation humidity. The conservation equation for species transport can be calculated as follows:

$$\frac{\partial(\rho C_{\mathrm{A}}^{*})}{\partial t} + \nabla \cdot \left(\rho C_{\mathrm{A}}^{*} \vec{v}\right) = \nabla \left[\left(\rho D + \frac{\mu_t}{Sc_t}\right)\nabla C_{\mathrm{A}}^{*}\right] + R_{\mathrm{A}} \tag{1}$$

where $C_{\mathrm{A}}^{*}$ is the local mass fraction of component A, $D$ is the mass diffusion coefficient for component A, $Sc_t$ is the turbulent Schmidt number, and $R_{\mathrm{A}}$ is the net production rate caused by the source term.

The boundary conditions for the numerical simulation calculation are set according to the actual working environment of a typical spring actuator; that is, the bottom surface of the air-fluid domain is set as an adiabatic non-slip wall surface, while the top surface and the four sides are set as opening conditions. The ambient temperature (233.15 K to 313.15 K), pressure (1 atm), and relative humidity (95%) are assigned to the side and top

surfaces of the external fluid domain. The heater portion in the solid domain is given a bulk heat source, and its body heat flux is set based on the heater power (10 W to 500 W). The remaining corresponding fluid surfaces and solid domain surfaces are set as fluid-solid coupling interfaces. Before setting up the calculation, initialization operations are required in all cases to speed up the calculation efficiency. The body heat source was assigned to the heater domain and the body heat flux was calculated according to the heating power. The side and top surfaces of the external fluid domain were set as open conditions.

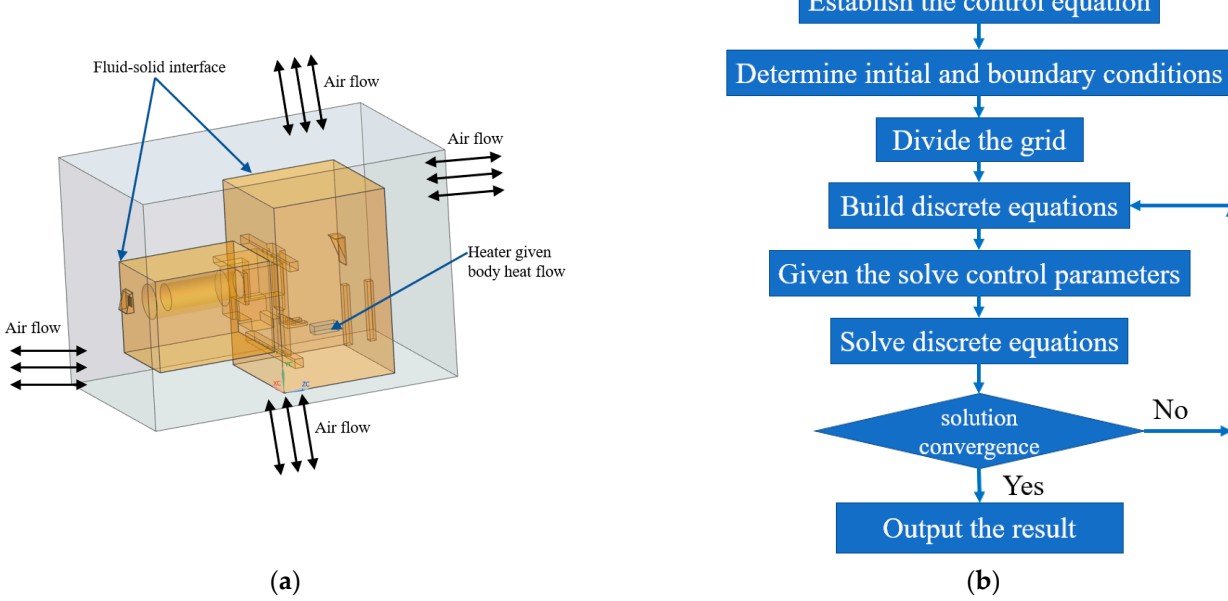

**Figure 2.** Numerical calculation model: (**a**) fluid-solid conjugate model; (**b**) calculation flowchart.

ICEM was used to perform the unstructured meshing of solids such as the typical spring actuator and components and the corresponding air fluids (as shown in Figure 3). The mesh of the solid domains and mesh of the fluid domains are given in Figure 3a,b, respectively. From Figure 3, it can be seen that the solid domain is composed entirely of tetrahedral mesh, while the fluid is composed of prismatic boundary layer mesh near the wall and tetrahedral mesh in the central region. From Figure 3b, it can be seen that the meshes of both the tiny structures and the near-wall region in the fluid domains are encrypted. The maximum size of the tetrahedral mesh for the solid and fluid central regions was set to 10 mm, and the mesh growth rate was set to 1.2. The first layer of the boundary layer mesh for the fluid was set to 0.01 mm, the number of layers was 15, and the mesh growth rate was set to 1.2. The total number of meshes was adjusted by adjusting the minimum size of the tetrahedral mesh. Table 2 shows the mesh-independence verification. In total, six sets of mesh models were completed for the numerical calculations, with total mesh numbers of 1.8 million, 2.6 million, 3.8 million, 5.0 million, 6.6 million, and 8.1 million, respectively. The mesh-independent validation results show that the minimum temperature and maximum relative humidity inside the chamber changed very little (within 2%) when the minimum tetrahedral mesh size was less than 0.5 mm, which indicates that the mesh-independent requirement is achieved when the total mesh number is 6.6 million. A meshing strategy with a minimum tetrahedral mesh size of 0.5 mm was used in all subsequent studies.

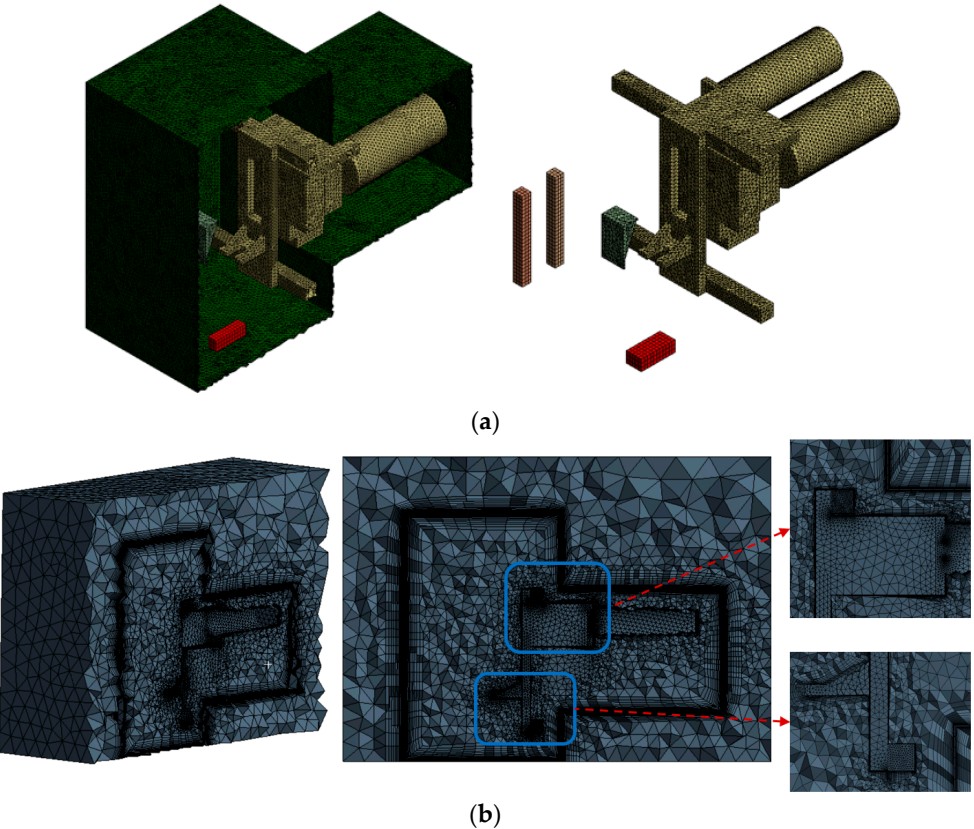

**(a)**

**(b)**

**Figure 3.** Mesh model of the typical spring actuator: (**a**) the mesh of the solid domains; (**b**) the mesh of the fluid domains.

**Table 2.** Mesh-independence verification.

| Fluid Domain | Solid Domain Mesh | Total Mesh | $RH_{max}$ | $T_{min}$/K |
|---|---|---|---|---|
| 1,600,000 | 200,000 | 1,800,000 | 50.4% | 314.2 |
| 2,200,000 | 400,000 | 2,600,000 | 55.6% | 309.7 |
| 3,200,000 | 600,000 | 3,800,000 | 57.9% | 306.9 |
| 4,200,000 | 800,000 | 5,000,000 | 59.5% | 305.1 |
| 5,600,000 | 1,000,000 | 6,600,000 | 61.1% | 303.8 |
| 7,000,000 | 1,200,000 | 8,200,000 | 61.5% | 304.6 |

The specific numerical method of validation is shown in Figure 4. The numerical method of this paper was validated by the experimental data. Figure 4a shows the experimental high- and low-temperature humidity and heat chamber of the TH-type model, which can provide a temperature range of −203–423 K and a relative humidity range of 10–98%, and can ensure temperature fluctuations of less than 0.5 K and humidity fluctuations of less than 5%. Temperature control is carried out inside the experimental chamber by generating high-temperature airflow for heating via different power heating parts and low-temperature airflow for cooling via different flow refrigerants. Humidity control is carried out through the combination of an electric steam humidifier and a dehumidification evaporator. Finally, the centrifugal fan in the experimental chamber is used to create circulating air so that the temperature and humidity distribution in the experimental chamber is uniform. The AM-type temperature and humidity sensor used for the humidity measurement is given in Figure 4b. The AM-type network-type humidity transmitter is a high-performance industrial temperature and humidity transmitter that measures the temperature and humidity of the environment in digital form and shows it on a local display. The measurement range of relative humidity is 5–100% and the accuracy is ± 3%.

Figure 4c,d shows the experimental measurements and experimental environment, respectively. Figure 4e,f gives the maximum temperature of the heater surface and the relative humidity in the chamber when the ambient temperature increases from 233 K to 313 K. The boundary conditions of the numerical model at the time of validation were consistent with the experimental model conditions, and the calculation of the numerical simulation was performed using Fluent software. The comparison results show that the temperature and humidity calculated by the numerical simulation in this paper deviated very little from the experimentally measured temperature and humidity. The deviation of the maximum heater surface temperature, $T_{heater}$, was −3.5–3.6 K, with a maximum deviation of 3.6 K. The relative deviation of the maximum relative humidity, $RH_{max}$, was −3.7–9.7%, with a maximum relative deviation of 9.7%. Therefore, the numerical simulation method used in this paper is accurate and reliable and can be used to numerically simulate the temperature and relative humidity fields of a typical spring actuator chamber.

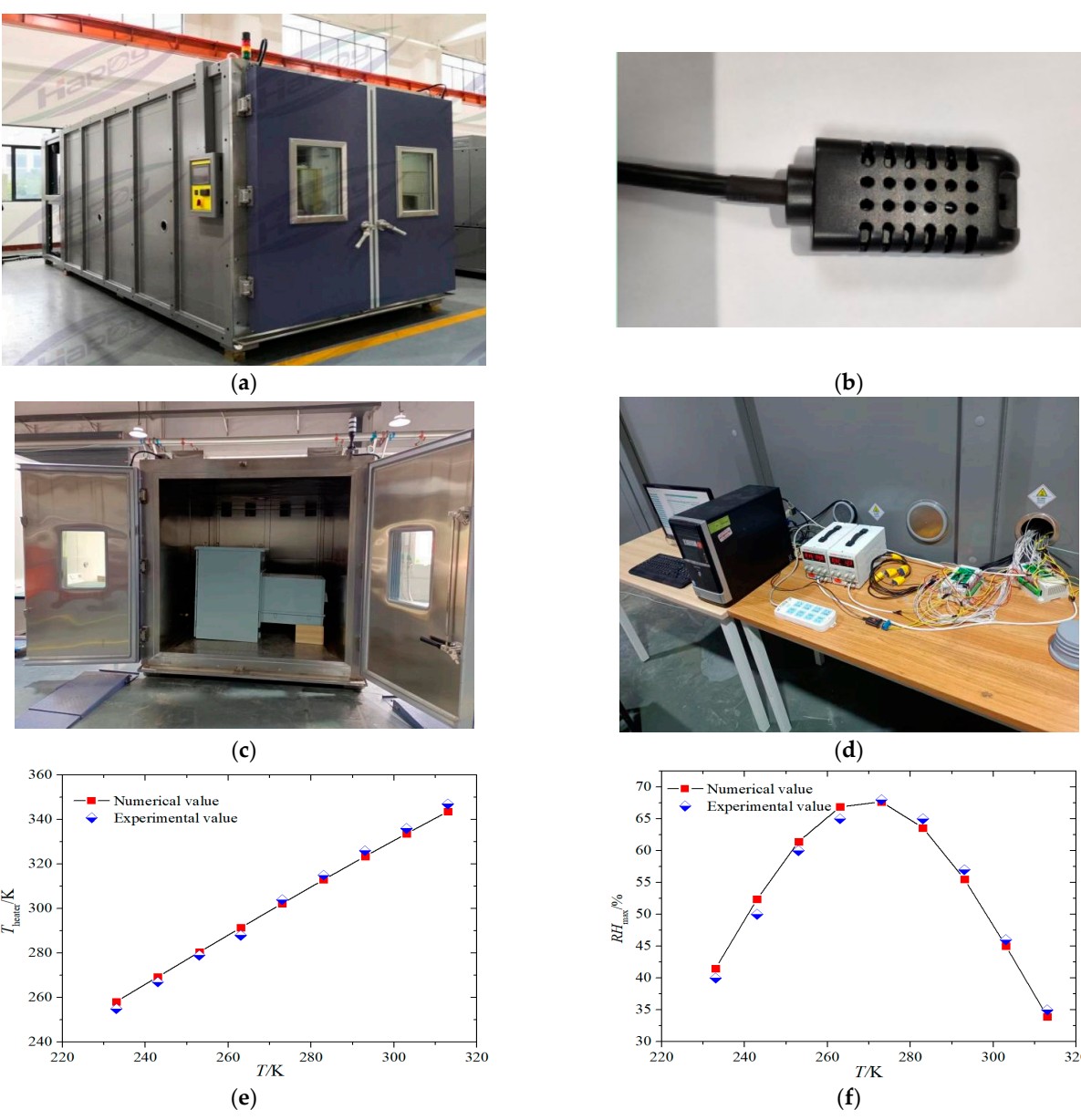

**Figure 4.** Numerical simulation method validation: (**a**) experiment chamber; (**b**) humidity sensor; (**c**) experimental environment; (**d**) experimental measurements; (**e**) numerical verification for $T_{heater}$; (**f**) numerical verification for $RH_{max}$.

### 3.2. Response Surface Model

The response surface design method is a widely used and effective method for constructing approximate models by fitting the relationship between response values and input parameters through polynomial functions. The advantages of the response surface design method are: (1) it is simple and convenient, needing fewer sample data; using simple polynomials can be more accurate in the local range to approximate most of the function relationships. (2) Robustness: second-order and third-order models can be chosen to fit complex nonlinear relationships. (3) High applicability: it can be applied widely in many engineering design fields.

Commonly used surrogate models include the response surface model, kriging model, radial basis function, neural network, etc. Compared with other approximation models, the response surface model offers the following advantages: it can approximate the function relationship more accurately in the local range with fewer trials, it can be shown by simple algebraic expressions, it can fit complex response relationships, and it has good robustness. The second-order response surface model is the most widely used and the fully quadratic response surface model can meet the fitting accuracy requirements of most nonlinear relationships [34]. Therefore, the second-order response surface model is used in this paper, and its expression is as follows [35]:

$$y = b_0 + \sum_{i=1}^{k} b_i x_i + \sum_{i=1}^{k} b_{ii} x_i^2 + \sum \sum_{i<j}^{k} b_{ij} x_i x_j + \varepsilon \tag{2}$$

where $b_0$ is the intercept; $b_i$, $b_{ii}$, $b_{ij}$ are the coefficients of the linear, square, and interaction terms; $x_i$, $x_j$ are the input parameters; $y$ is the response value; $\varepsilon$ is the error term.

In order to fit a response surface model with high accuracy using fewer sample data, a reasonable experimental design method is needed to obtain the sample data. The central composite design is one of the most commonly used experimental design methods for the fitting second-order response surface model [36], which has the advantages of a small number of cases to be run and a simple design; its structure is shown in Figure 5. Therefore, this paper adopts the central composite design method to arrange the cases for numerical simulation calculation, while the specific design operations are completed using Minitab (2018, Minitab Inc., State College, PA, USA) 18.0. In addition, Figure 5 shows a central composite design in a three-dimensional space (i.e., three factors). The whole test consists of three main test points, including the corner points (blue triangles in the figure, with a number of $2^k$), the center points, and the axis points (red squares in the figure, with a number of $2k$). The first five columns in Table 2 give the specific design table.

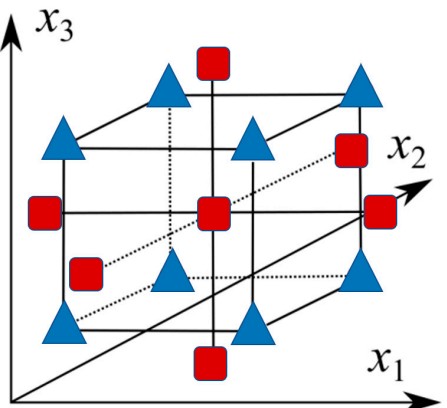

**Figure 5.** The center composite design.

### 3.3. Optimization of Design Methods

As the typical spring actuator works in different positions and seasons, the environmental temperature and relative humidity are different, the environmental temperature range is 233–313 K, and the environmental relative humidity can reach 95% in most cases. In order to ensure that the typical spring actuator can operate normally under different environmental temperatures and relative humidities, the temperature inside the chamber needs to be kept above 265 K and the relative humidity below 85%. A heater is usually used to warm the air inside the spring actuator to ensure the temperature and relative humidity environment inside the chamber, but the higher heating power may also lead to high temperatures on the heater surface. To ensure that the heater can operate at a safe temperature below 340 K, the heater size needs to be changed to reduce the heater surface temperature. Therefore, optimal designing of the heater power and size needs to be carried out.

Based on the above requirements, the following multi-objective optimization problem is established for different environmental temperatures.

Optimization Objective:

$T_{\min} \geq 265 \text{ K}$
$RH_{\max} \leq 85\%$
$T_{\text{heater}} \leq 340 \text{ K}.$

Constraints:

$10 \text{ W} \leq Q \leq 500 \text{ W}$
$2 \leq L/H \leq 6$
$1 \leq W/H \leq 6.5.$

A genetic algorithm (GA) was used to carry out the optimal designing of the heater size inside the chamber of a typical spring actuator. The flow chart of the genetic algorithm is shown in Figure 6 and its optimization process includes: initialization of the population; calculation of the fitness function and taking of the optimal value; judgment as to whether the optimization conditions are satisfied; the selection operation if the conditions are not satisfied; the crossover operation; the variation operation; cycling through the previous steps until the optimization conditions are satisfied and the final optimal value is obtained.

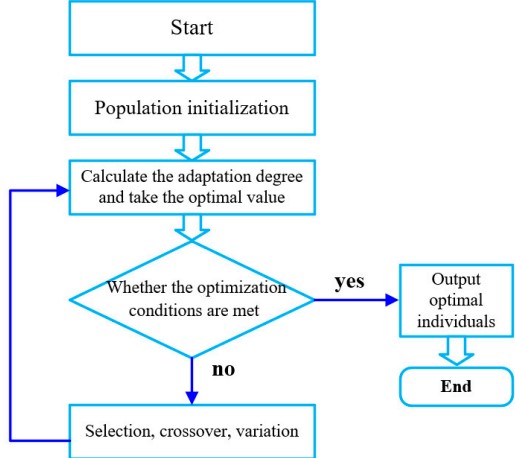

**Figure 6.** The flow chart of the genetic algorithm.

## 4. Analysis of the Results and Discussion

It has seldom been demonstrated that the temperature and humidity of a spring actuator can be controlled by a heater system and that it can be operated safely at different environmental temperatures. This study is unique in that: (1) the temperature and humidity distribution inside the spring actuator is demonstrated; (2) the significance of the research parameters with different responses is explored; (3) an explicit empirical formula with high

fitting accuracy is given; (4) the parameters of the heater at different ambient temperatures are optimally obtained.

### 4.1. Temperature and Relative Humidity Field Distribution Inside the Chamber

Figure 7a,b gives the flow situation and temperature and relative humidity distribution inside a typical spring actuator chamber with an environmental temperature of 293 K, pressure of 1 atm, relative humidity of 95%, and a heating power of 100 W. As can be seen from Figure 7, the air around the heater starts to move upward after being heated. Since the heater is in the lower center of the large chamber on the right, the air flows to the large chamber after the top wall, creating a scattered flow in all directions, forming a petal-like flow field after passing through the small chamber on the left and out of the louvered outlet. The surface of the heater has the highest temperature and lowest relative humidity, resulting in high air temperature and low relative humidity around and directly above the heater. The air then starts to flow from the top of the large chamber on the right side in the direction of the length and width of the chamber and gradually flows down through the parts inside the small chamber. The temperature of the air gradually decreases, and the relative humidity gradually increases during this process, resulting in the air around the other parts inside the chamber showing a trend of gradually decreasing temperature and gradually increasing relative humidity from top to bottom. In addition, it can be seen in Figure 7 that the temperature of the heater surface far exceeds the temperature of the other parts, which is due to the small size of the heater on the one hand and the large heating power set on the other hand, which does not allow the heat on the heater surface to be distributed quickly under natural convection, resulting in a high local temperature on the heater surface. Therefore, the size and power of the typical spring actuator heater need to be optimally designed to meet the operating requirements of the spring actuator.

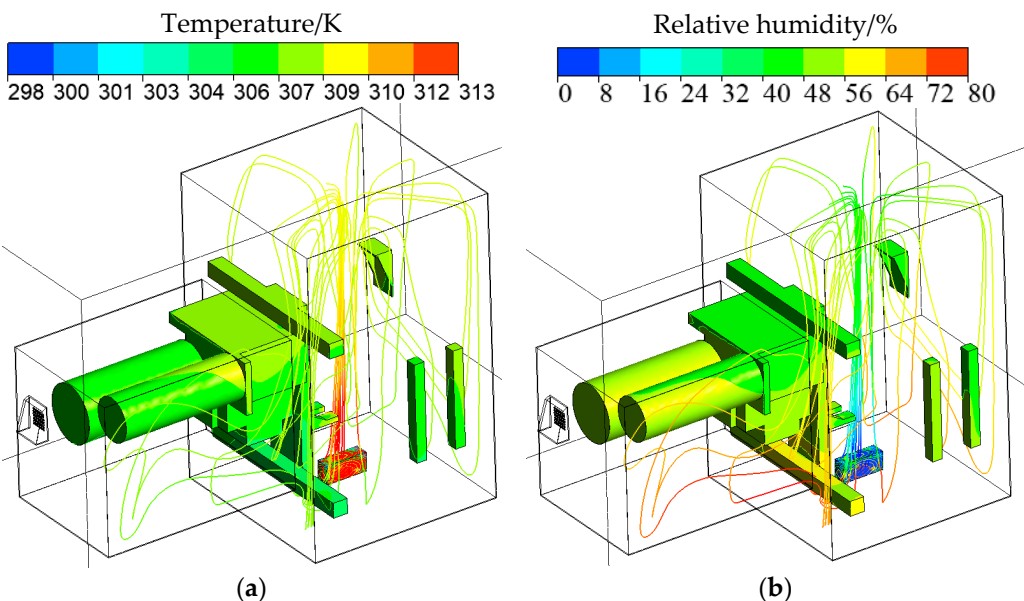

**Figure 7.** Flow situation, temperature, and relative humidity distribution in the actuator chamber: (**a**) solid surface temperature distribution; (**b**) relative humidity distribution around the solid surface.

### 4.2. Response Surface Model Evaluation

The arrangement and calculation results of the research parameters obtained, based on the central composite design, are shown in Table 3, where the response values were obtained via a Fluent numerical simulation. The input parameters include the environmental temperature $T$, heater power $Q$, heater relative length $L/H$, and heater relative width $W/H$. The response values include the minimum temperature inside the chamber, $T_{\min}$, the maximum relative humidity, $RH_{\max}$, and the maximum temperature of the heater surface,

$T_{\text{heater}}$. The four input parameters have low, medium, and high values; therefore, a total of 31 numerical simulation cases are arranged for the central composite design in this paper. The order of the cases is randomly generated using Minitab.

**Table 3.** Arrangement of the research parameters and calculation results.

| Order | *T* (K) | *Q* (W) | *L/H* | *W/H* | $T_{\text{min}}$ (K) | $RH_{\text{max}}$ (%) | $T_{\text{heater}}$ (K) |
|---|---|---|---|---|---|---|---|
| 1 | 313 | 500 | 2 | 5 | 353.024 | 2.679 | 485.133 |
| 2 | 233 | 500 | 6 | 1 | 259.734 | 8.176 | 362.398 |
| 3 | 233 | 500 | 2 | 1 | 261.315 | 7.185 | 535.017 |
| 4 | 233 | 500 | 6 | 5 | 260.766 | 7.515 | 321.384 |
| 5 | 273 | 255 | 4 | 3 | 292.781 | 28.416 | 354.459 |
| 6 | 273 | 255 | 2 | 3 | 293.364 | 27.095 | 390.602 |
| 7 | 273 | 255 | 4 | 3 | 292.781 | 28.416 | 354.459 |
| 8 | 273 | 255 | 4 | 1 | 292.6 | 28.84 | 381.545 |
| 9 | 273 | 10 | 4 | 3 | 274.285 | 96.777 | 279.458 |
| 10 | 233 | 10 | 6 | 1 | 234.024 | 66.8 | 238.578 |
| 11 | 233 | 255 | 4 | 3 | 248.614 | 20.28 | 304.417 |
| 12 | 273 | 255 | 6 | 3 | 292.376 | 29.374 | 340.227 |
| 13 | 313 | 500 | 6 | 1 | 351.213 | 3.107 | 473.282 |
| 14 | 313 | 10 | 2 | 1 | 313.901 | 65.496 | 324.64 |
| 15 | 273 | 255 | 4 | 5 | 292.478 | 29.13 | 340.921 |
| 16 | 273 | 255 | 4 | 3 | 292.781 | 28.416 | 354.459 |
| 17 | 313 | 10 | 6 | 5 | 314.379 | 62.985 | 318.091 |
| 18 | 273 | 255 | 4 | 3 | 292.781 | 28.416 | 354.459 |
| 19 | 313 | 500 | 6 | 5 | 351.119 | 3.131 | 425.213 |
| 20 | 233 | 10 | 2 | 5 | 234.143 | 66.153 | 238.879 |
| 21 | 233 | 500 | 2 | 5 | 261.101 | 7.312 | 366.53 |
| 22 | 313 | 500 | 2 | 1 | 349.982 | 3.435 | 649.303 |
| 23 | 233 | 10 | 2 | 1 | 234.031 | 66.761 | 244.23 |
| 24 | 273 | 255 | 4 | 3 | 292.781 | 28.416 | 354.459 |
| 25 | 273 | 500 | 4 | 3 | 309.145 | 7.464 | 418.564 |
| 26 | 273 | 255 | 4 | 3 | 292.781 | 28.416 | 354.459 |
| 27 | 273 | 255 | 4 | 3 | 292.781 | 28.416 | 354.459 |
| 28 | 233 | 10 | 6 | 5 | 234.081 | 66.488 | 237.02 |
| 29 | 313 | 10 | 6 | 1 | 314.057 | 64.666 | 319.821 |
| 30 | 313 | 255 | 4 | 3 | 334.577 | 12.094 | 403.291 |
| 31 | 313 | 10 | 2 | 5 | 314.316 | 63.313 | 320.301 |

To fit the correlation equation more accurately, the responses were optimally transformed. The Box–Cox transformation is a commonly used tool for data analysis [37] and has two main objectives; the first is that the Box–Cox transformation can reduce the unobservable error and the correlation of the predictor variables to some extent. The second is to use this transformation to make the dependent variable distribution into normal distribution or make it more stable. Therefore, in this paper, the Box–Cox transformation was applied to the three responses, which can make the fitted formulas more accurate. The minimum temperature in the chamber, $T_{\text{min}}$, was transformed to $T_{\text{min}}{}^{0.5}$, the maximum relative humidity, $RH_{\text{max}}$, was transformed to Ln ($RH_{\text{max}}$), and the maximum heater surface temperature, $T_{\text{heater}}$, was transformed to Ln ($T_{\text{heater}}$). The coefficients of the fitted correlations when the responses were $T_{\text{min}}{}^{0.5}$, Ln ($RH_{\text{max}}$), and Ln ($T_{\text{heater}}$), respectively, are given in Table 4. As can be seen from Table 3, there are 15 coefficients in the empirical formulae obtained from the experimental design and the response surface method fitting, of which one is a real term, four are linear terms, four are square terms, and six are interaction terms. The specific empirical formulas are detailed in Equation (2) of Section 3.2.

**Table 4.** Arrangement of the research parameters and calculation results.

| Coefficient | $T_{\min}{}^{0.5}$ | Ln ($RH_{\max}$) | Ln ($T_{\text{heater}}$) |
|---|---|---|---|
| $b_0$ | 4.477 | −22.68 | 4.286 |
| $b_1$ | 0.05845 | 0.2016 | 0.00813 |
| $b_2$ | 0.00109 | −0.000218 | 0.00231 |
| $b_3$ | −0.0096 | 0.0201 | −0.0919 |
| $b_4$ | 0.0032 | −0.0061 | −0.0757 |
| $b_{11}$ | −0.000052 | −0.000368 | −0.000009 |
| $b_{22}$ | −0.000001 | 0.000002 | −0.000001 |
| $b_{33}$ | 0.00013 | −0.0001 | 0.00625 |
| $b_{44}$ | −0.00228 | 0.00666 | 0.00358 |
| $b_{12}$ | 0.000004 | −0.000022 | −0.000001 |
| $b_{13}$ | 0.000039 | −0.000097 | 0.000064 |
| $b_{14}$ | 0.000054 | −0.000172 | 0.000086 |
| $b_{23}$ | −0.00001 | 0.000029 | −0.000117 |
| $b_{24}$ | 0.00001 | −0.00003 | −0.000108 |
| $b_{34}$ | −0.00077 | 0.0026 | 0.00728 |

In Tables 5–7, the degree of freedom (DOF) is equal to the number of levels of the variable minus 1. As can be seen from the tables, there is one real term, four linear terms, four squared terms, and six interaction terms, which add up to 15 variables. In the ANOVA, a large fluctuation in the research parameter indicates a significant effect on the response. *Adj.SS* stands for the adjusted sum of squares deviation, which indicates, to some extent, the magnitude of fluctuation in the data. *Adj.MS* stands for adjusted mean squared deviation, which is the correction of *Adj.MS* based on the amount of data, where *Adj.MS = Adj.SS/DOF*. The F and *p*-values are statistics used in the analysis of variance (ANOVA) for hypothesis testing. A larger F value indicates that a term is more significant. When the *p*-value is less than 0.05, it means that the term is significant to the model. When the *p*-value is greater than 0.05, this means that the term is negligible. More details of the ANOVA are recorded elsewhere in the literature [38].

**Table 5.** The results of the ANOVA for $T_{\min}$.

| Source | DOF | Adj.SS | Adj.MS | F Value | p-Value |
|---|---|---|---|---|---|
| Regression | 14 | 32.5579 | 2.3256 | 6235.16 | 0.000 |
| Linear | 4 | 32.4117 | 8.1029 | 21,725.04 | 0.000 |
| $T$ | 1 | 28.4067 | 28.4067 | 76,162.30 | 0.000 |
| $Q$ | 1 | 4.0035 | 4.0035 | 10,733.89 | 0.000 |
| $L/H$ | 1 | 0.0006 | 0.0006 | 1.56 | 0.230 |
| $W/H$ | 1 | 0.0009 | 0.0009 | 2.39 | 0.141 |
| Square | 4 | 0.1154 | 0.0289 | 77.38 | 0.000 |
| $T \times T$ | 1 | 0.0181 | 0.0181 | 48.44 | 0.000 |
| $Q \times Q$ | 1 | 0.0043 | 0.0043 | 11.62 | 0.004 |
| $L/H \times L/H$ | 1 | 0.0000 | 0.0000 | 0.00 | 0.965 |
| $W/H \times W/H$ | 1 | 0.0002 | 0.0002 | 0.58 | 0.457 |
| Interaction | 6 | 0.0308 | 0.0051 | 13.76 | 0.000 |
| $T \times Q$ | 1 | 0.0294 | 0.0294 | 78.85 | 0.000 |
| $T \times L/H$ | 1 | 0.0002 | 0.0002 | 0.42 | 0.528 |
| $T \times W/H$ | 1 | 0.0003 | 0.0003 | 0.79 | 0.388 |
| $Q \times L/H$ | 1 | 0.0004 | 0.0004 | 1.10 | 0.309 |
| $Q \times W/H$ | 1 | 0.0004 | 0.0004 | 1.01 | 0.330 |
| $L/H \times W/H$ | 1 | 0.0002 | 0.0002 | 0.41 | 0.531 |
| Residual error | 16 | 0.0060 | 0.0004 | - | - |
| Lack-of-fit | 10 | 0.0060 | 0.0006 | - | - |
| Pure error | 6 | 0.0000 | 0.0000 | - | - |
| Total | 30 | 32.5639 | - | - | - |

**Table 6.** The results of the ANOVA for $RH_{\max}$.

| Source | DOF | Adj.SS | Adj.MS | F Value | p-Value |
|---|---|---|---|---|---|
| Regression | 14 | 34.9241 | 2.4946 | 821.98 | 0.000 |
| Linear | 4 | 32.2422 | 8.0605 | 2656.01 | 0.000 |
| $T$ | 1 | 1.0043 | 1.0043 | 330.91 | 0.000 |
| $Q$ | 1 | 31.2259 | 31.2259 | 10,289.14 | 0.000 |
| $L/H$ | 1 | 0.0044 | 0.0044 | 1.44 | 0.247 |
| $W/H$ | 1 | 0.0077 | 0.0077 | 2.53 | 0.131 |
| Square | 4 | 1.9320 | 0.4830 | 159.15 | 0.000 |
| $T \times T$ | 1 | 0.9002 | 0.9002 | 296.61 | 0.000 |
| $Q \times Q$ | 1 | 0.0229 | 0.0229 | 7.55 | 0.014 |
| $L/H \times L/H$ | 1 | 0.0000 | 0.0000 | 0.00 | 0.991 |
| $W/H \times W/H$ | 1 | 0.0018 | 0.0018 | 0.61 | 0.447 |
| Interaction | 6 | 0.7500 | 0.1250 | 41.19 | 0.000 |
| $T \times Q$ | 1 | 0.7377 | 0.7377 | 243.07 | 0.000 |
| $T \times L/H$ | 1 | 0.0010 | 0.0010 | 0.32 | 0.580 |
| $T \times W/H$ | 1 | 0.0030 | 0.0030 | 1.00 | 0.332 |
| $Q \times L/H$ | 1 | 0.0031 | 0.0031 | 1.04 | 0.324 |
| $Q \times W/H$ | 1 | 0.0034 | 0.0034 | 1.12 | 0.305 |
| $L/H \times W/H$ | 1 | 0.0017 | 0.0017 | 0.57 | 0.461 |
| Residual error | 16 | 0.0486 | 0.0030 | - | - |
| Lack-of-fit | 10 | 0.0486 | 0.0049 | - | - |
| Pure error | 6 | 0.0000 | 0.0000 | - | - |
| Total | 30 | 34.9727 | - | - | - |

**Table 7.** The results of the ANOVA for $T_{\text{heater}}$.

| Source | DOF | Adj.SS | Adj.MS | F Value | p-Value |
|---|---|---|---|---|---|
| Regression | 14 | 1.54213 | 0.110152 | 123.88 | 0.000 |
| Linear | 4 | 1.42236 | 0.355591 | 399.91 | 0.000 |
| $T$ | 1 | 0.33819 | 0.338191 | 380.35 | 0.000 |
| $Q$ | 1 | 0.94731 | 0.947310 | 1065.39 | 0.000 |
| $L/H$ | 1 | 0.07478 | 0.074784 | 84.11 | 0.000 |
| $W/H$ | 1 | 0.06208 | 0.062077 | 69.81 | 0.000 |
| Square | 4 | 0.00620 | 0.001550 | 1.74 | 0.190 |
| $T \times T$ | 1 | 0.00055 | 0.000554 | 0.62 | 0.442 |
| $Q \times Q$ | 1 | 0.00391 | 0.003906 | 4.39 | 0.052 |
| $L/H \times L/H$ | 1 | 0.00162 | 0.001624 | 1.83 | 0.195 |
| $W/H \times W/H$ | 1 | 0.00053 | 0.000531 | 0.60 | 0.451 |
| Interaction | 6 | 0.11357 | 0.018929 | 21.29 | 0.000 |
| $T \times Q$ | 1 | 0.00130 | 0.001302 | 1.46 | 0.244 |
| $T \times L/H$ | 1 | 0.00042 | 0.000423 | 0.48 | 0.500 |
| $T \times W/H$ | 1 | 0.00075 | 0.000750 | 0.84 | 0.372 |
| $Q \times L/H$ | 1 | 0.05243 | 0.052433 | 58.97 | 0.000 |
| $Q \times W/H$ | 1 | 0.04508 | 0.045084 | 50.70 | 0.000 |
| $L/H \times W/H$ | 1 | 0.01358 | 0.013579 | 15.27 | 0.001 |
| Residual error | 16 | 0.01423 | 0.000889 | - | - |
| Lack-of-fit | 10 | 0.01423 | 0.001423 | - | - |
| Pure error | 6 | 0.00000 | 0.000000 | - | - |
| Total | 30 | 1.55636 | - | - | - |

The comparison in Figure 8 gives the numerically calculated values and the predicted values of RSM for a typical spring actuator. Figure 8a–c shows a comparison of the two values when the response is the minimum temperature inside the chamber, $T_{\min}$, the maximum relative humidity, $RH_{\max}$, and the maximum temperature of the heater, $T_{\text{heater}}$, respectively. In the figure, the solid black line indicates the numerically calculated value, while the red scatter indicates the RSM predicted value, and the blue line indicates the $\pm 1\%$, $\pm 10\%$, or $\pm 5\%$ deviation of the numerically calculated value. As can be seen

in Figure 8, the RSM predictions for the minimum temperature in the chamber are all distributed within $\pm 1\%$ of the numerically calculated value, and the RSM predictions for the maximum relative humidity are basically distributed within $\pm 10\%$ of the numerically calculated value, while the RSM predictions for the maximum temperature of the heater are all distributed within $\pm 5\%$ of the numerically calculated value. For the correlation of the minimum temperature inside the chamber, the maximum deviation is 0.52% and the average absolute deviation is 0.11%. For the correlation of maximum relative humidity, the maximum deviation is 11.73% and the average absolute deviation is 2.72%. For the correlation of maximum heater temperature, the maximum deviation is 4.71% and the average absolute deviation is 1.63%. The root mean square error (RMSE), the coefficient of determination $R^2$, and $R^2$ (adjusted) of the empirical equation obtained by fitting the response surface model can be calculated; the corresponding formulae are detailed in the literature [39], and the results are shown in Table 8. As can be seen from Table 8, when the responses match the minimum temperature, maximum relative humidity, and maximum temperature of the heater inside the chamber, respectively, the root mean square error (RMSE) of the empirical formula are 0.0193, 0.0551, and 0.0298, all of which are less than 0.06; the coefficients of determination $R^2$ of the empirical formula are 99.98%, 99.86%, and 99.09%, all of which are greater than 99%. The coefficients of determination $R^2$ (adjusted) are 99.97%, 99.74%, and 98.29%, respectively, all of which are greater than 98%. The above results show that the second-order response surface model fitted in this paper can be used to predict the minimum temperature, maximum relative humidity, and maximum temperature of the heater surface in a typical spring actuator. It also provides an accurate approximate model for the optimal design of a typical spring actuator heater.

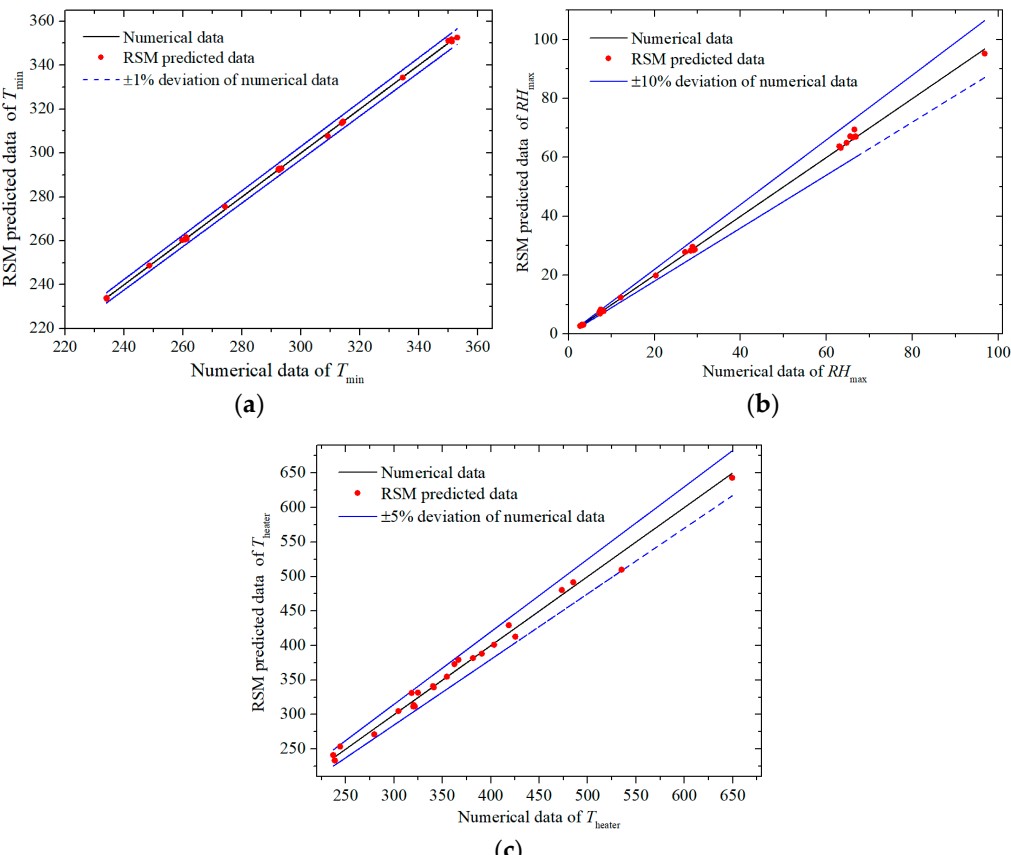

**Figure 8.** The deviation of correlation fitting: (**a**) $T_{\min}$; (**b**) $RH_{\max}$; (**c**) $T_{\text{heater}}$.

**Table 8.** Fitting accuracy of the response surface model.

| Evaluation Index | $T_{\text{min}}$ | $RH_{\text{max}}$ | $T_{\text{heater}}$ |
|---|---|---|---|
| RMSE | 0.0193 | 0.0551 | 0.0298 |
| $R^2$ | 99.98% | 99.86% | 99.09% |
| $R^2$ (Adjusted) | 99.97% | 99.74% | 98.29% |

*4.3. Significance Analysis of the Research Parameters*

The significance of the input parameters ($T$, $Q$, $L/H$, and $W/H$) for different responses ($T_{\text{min}}$, $RH_{\text{max}}$, and $T_{\text{heater}}$) was analyzed using the central composite design (CCD)-response surface method (RSM). Figures 9–11 give the Pareto effect plots and normal effect plots. In these figures, the standardized effect is the dimensionless value and the importance of the research parameters under a certain response; a larger value indicates that the effect on the response is greater. $A$, $B$, $C$, and $D$ represent the research parameters $T$, $Q$, $L/H$, and $W/H$, respectively. $A$–$D$ represents the effect of the linear term of the research parameters on the response, $AB$–$CD$ represents the effect of the interaction term of the research parameters on the response, and $AA$–$DD$ represents the effect of the squared term of the research parameters on the response. In the Pareto effect plot, the positive effects are distributed to the right of the red line and the negative effects are distributed to the left of the red line. In the normal effects plot, the effects of each factor are arranged in a sequence from small to large (positive and negative signs are taken into account), and these effect points are labeled on the normal probability plot. As can be seen from Figure 9, the effective terms of $T_{\text{min}}$, ranked from highest to lowest sensitivity levels, are $T$, $Q$, $T \times Q$, $T \times T$, and $Q \times Q$, where $T$, $Q$, and $T \times Q$ have positive effects on $T_{\text{min}}$, while $T \times T$ and $Q \times Q$ have negative effects on $T_{\text{min}}$. Similarly, as shown in Figure 10, the effective terms of $RH_{\text{max}}$, sorted from high to low sensitivity levels, are $Q$, $T$, $T \times T$, $T \times Q$, and $Q \times Q$, where $Q \times Q$ has a positive effect on $RH_{\text{max}}$ and $Q$, $T$, $T \times T$, and $T \times Q$ have a negative effect on $RH_{\text{max}}$. Correspondingly, as shown in Figure 11, the effective terms of $T_{\text{heater}}$, sorted from high to low sensitivity levels, are $T$, $Q$, $L/H$, $W/H$, $Q \times L/H$, $Q \times W/H$, and $L/H \times W/H$, where $Q$, $T$, and $L/H \times W/H$ have positive effects on $T_{\text{heater}}$, while $L/H$, $W/H$, $Q \times L/H$, and $Q \times W/H$ have negative effects on $T_{\text{heater}}$. In summary, among the four influencing parameters of $T$, $Q$, $L/H$, and $W/H$, $T$ is the most significant parameter affecting $T_{\text{min}}$, followed by $Q$, $W/H$, and $L/H$; $Q$ has the most significant effect on $RH_{\text{max}}$, followed by $T$, $W/H$, and $L/H$; the most significant effect on $T_{\text{heater}}$ is $Q$, followed by $T$, $L/H$, and $W/H$.

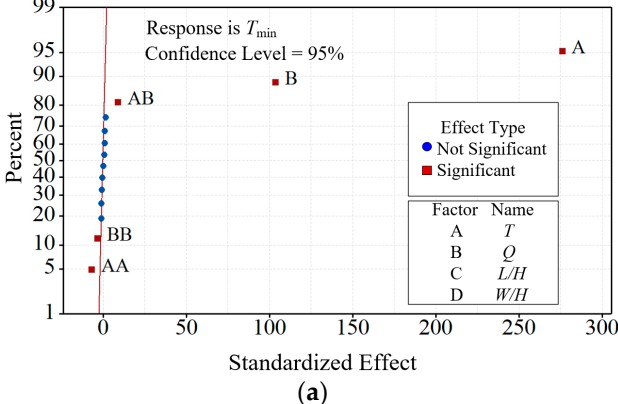

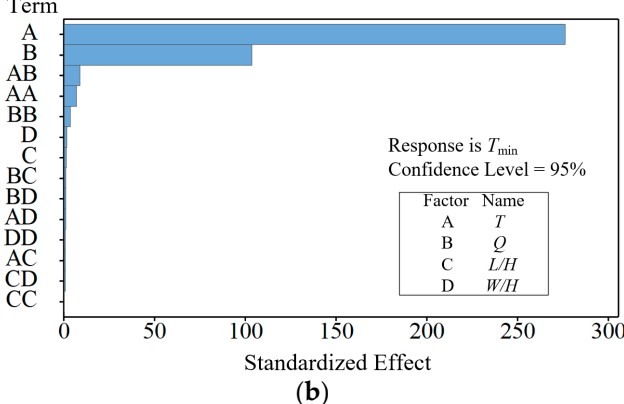

**Figure 9.** The factor effect diagram of $T_{\text{min}}$: (**a**) the Pareto effect diagram; (**b**) the normal effect diagram.

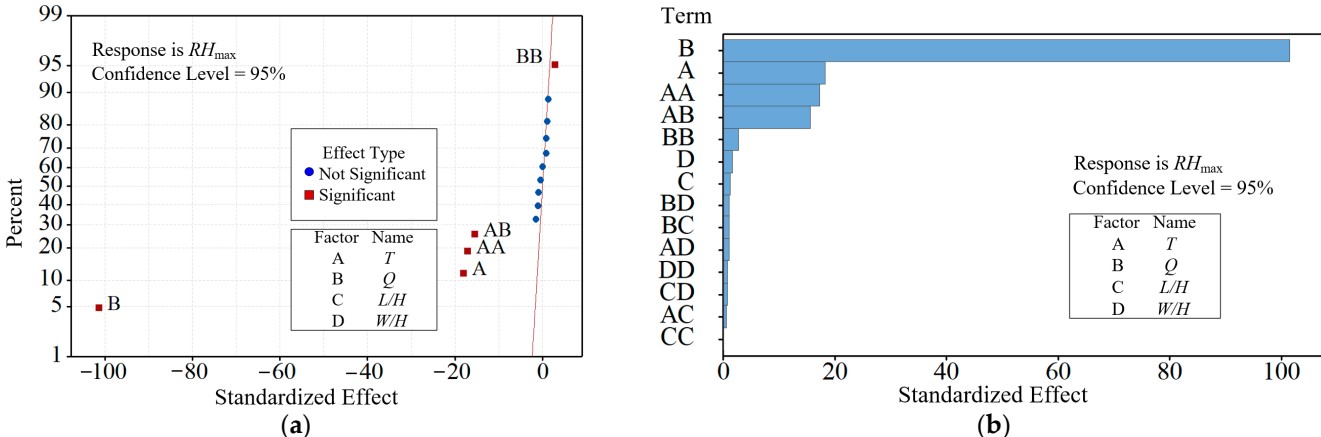

**Figure 10.** The factor effect diagram of $RH_{max}$: (**a**) the Pareto effect diagram; (**b**) the normal effect diagram.

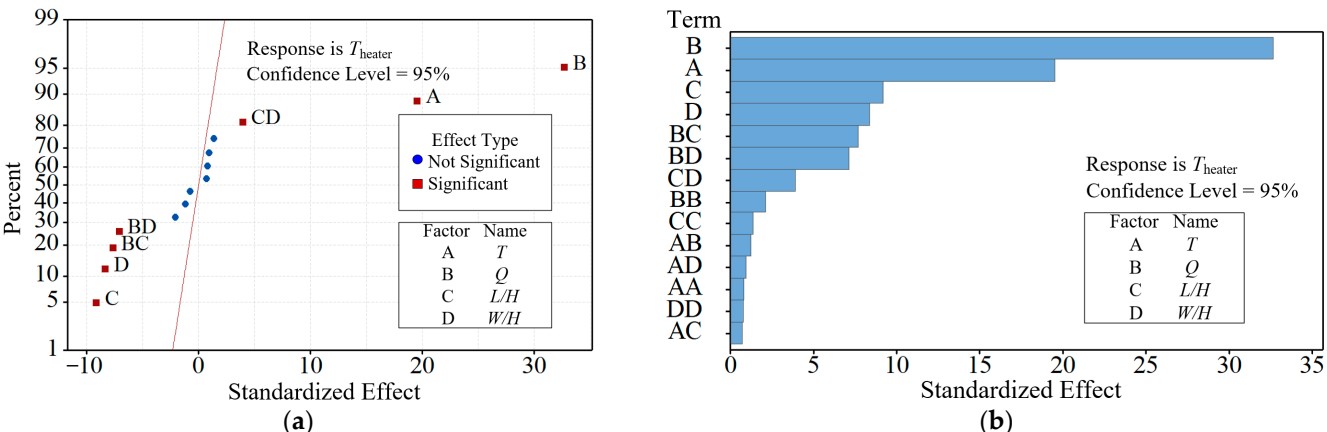

**Figure 11.** The factor effect diagram of $T_{heater}$: (**a**) Pareto effect diagram; (**b**) normal effect diagram.

### 4.4. Analysis of Parameter Optimization Results

According to the characteristics of the sample data obtained from the numerical simulation in Table 2 of this paper, the appropriate parameters of the genetic optimization algorithm were selected. Specifically, the population size was selected as 10, the crossover probability was selected as 0.3, the variation probability was selected as 0.2, and the number of genetic generations was set as 1000. The specific settings are detailed in the literature [40].

Based on the genetic algorithm-based optimization algorithm and its constraints, as established in Section 3.3, the optimal designing of the typical spring actuator heater size was carried out, and the optimal size of the typical spring actuator heater for different environmental temperatures was obtained. The optimization results are listed in Table 9. From Table 9, it can be seen that the optimal value of heater power is kept as 10 W when the environmental temperature is reduced from 313 K to 283 K, i.e., a heating power of 10 W can make the minimum temperature inside the spring actuator chamber that was used in this paper greater than 265 K and the maximum relative humidity less than 85%. To achieve a heater surface temperature lower than 340 K, the relative length $L/H$ of the heater was slightly reduced (from 3.57 to 3.15) and the relative width $W/H$ was kept at 1. This is because the environmental temperature was lower when the setup was more conducive to heat dissipation on the heater surface so that the heater heat-dissipation area (i.e., heater size) was smaller when the temperature was lower under the same heating power. Subsequently, as the environmental temperature continued to decrease, the heater power increased rapidly, and the optimal relative length and the optimal relative width of the heater also increased rapidly to reduce the heater surface area. When the environmental

temperature was 273 K, the optimal value of heater power was 50 W, the corresponding optimal relative length of the heater was 3.88, and the optimal relative width was 1. When the environmental temperature was 263 K, the optimal value of heater power was 200 W, the corresponding optimal relative length of the heater was 3.86, and the optimal relative width was 3.9. When the environmental temperature decreased to 243 K and 233 K, the optimal value of heater power increased to 420 W and 490 W, respectively, the corresponding optimal relative length of the heater increased to 6, and the optimal relative width increased to 3.7 and 5.3, respectively. In addition, it can also be seen from Table 9 that the minimum temperature in the actuator chamber, $T_{\min}$, the maximum relative humidity, $RH_{\max}$, and the maximum temperature on the surface of the heater, $T_{\text{heater}}$, obtained from the optimization met the optimization requirements.

**Table 9.** Optimization results.

| Temperature $T$ | Optimal $Q$ | Optimal $L/H$ | Optimal $W/H$ | $T_{\min}$ | $RH_{\max}$ | $T_{\text{heater}}$ |
|---|---|---|---|---|---|---|
| 313 | 10 | 3.57 | 1 | 313.731 | 66.318 | 316.158 |
| 303 | 10 | 3.45 | 1 | 304.384 | 82.014 | 308.210 |
| 293 | 10 | 3.31 | 1 | 294.821 | 83.629 | 300.167 |
| 283 | 10 | 3.15 | 1 | 285.061 | 75.971 | 292.088 |
| 273 | 50 | 3.88 | 1 | 278.101 | 84.360 | 294.220 |
| 263 | 200 | 3.86 | 3.9 | 278.483 | 38.559 | 319.521 |
| 253 | 380 | 3.6 | 6.5 | 277.884 | 15.759 | 332.539 |
| 243 | 420 | 6 | 3.7 | 268.615 | 11.990 | 329.531 |
| 233 | 490 | 6 | 5.3 | 260.032 | 8.002 | 308.674 |

## 5. Conclusions

In this paper, the temperature and relative humidity calculation and heater optimization design study of a typical spring actuator heating system was carried out, based on numerical simulation methods. The study can provide a reference for the temperature and humidity control system for future actuators. The following main conclusions were obtained:

(1) The temperature of the heater surface far exceeded the temperature of other parts. The air around the other parts showed a trend of gradually decreasing temperature and increasing relative humidity from top to bottom.

(2) The maximum prediction errors of the fitted second-order response surface model for the minimum temperature, maximum relative humidity, and maximum temperature of the heater surface inside the typical spring chamber were 0.5%, 11.7%, and 4.7%, respectively.

(3) Among the four influencing parameters of $T$, $Q$, $L/H$, and $W/H$, $T$ is the most significant parameter affecting $T_{\min}$, followed by $Q$, $W/H$, and $L/H$; $Q$ has the most significant effect on $RH_{\max}$, followed by $T$, $W/H$, and $L/H$; the most significant effect on $T_{\text{heater}}$ is $Q$, followed by $T$, $L/H$, and $W/H$.

(4) The optimal heating power of the heater increased from 10 W to 490 W, the optimal relative length increased from 3.57 to 6, and the optimal relative width increased from 1 to 5.3 when the environmental temperature was reduced from 313 K to 233 K.

**Author Contributions:** Conceptualization, Z.Z. and L.X. (Lei Xi); methodology, Z.Z. and L.X. (Liang Xu); software, Z.Z.; validation, Z.Z., L.X. (Liang Xu); formal analysis, Z.Z. and J.G.; investigation, Z.Z. and Y.L.; resources, Z.Z., L.X. (Lei Xi) and J.G.; data curation, Z.Z. and Y.L.; writing—original draft preparation, Z.Z. and L.X. (Liang Xu); writing—review and editing, L.X. (Liang Xu), Z.Z. and L.X. (Lei Xi); visualization, L.X. (Lei Xi) and Z.Z.; supervision, L.X. (Liang Xu) and J.G.; project administration, J.G.; funding acquisition, L.X. (Lei Xi). All authors have read and agreed to the published version of the manuscript.

**Funding:** This research was funded by the project supported by the Natural Science Basic Research Plan in Shaanxi Province of China (2022JQ-545), the project funded by the China Postdoctoral Science Foundation (2021M702573), and the National Natural Science Foundation of China (51876157).

**Data Availability Statement:** Not applicable.

**Conflicts of Interest:** The authors declare no conflict of interest.

## Nomenclature

| | |
|---|---|
| ANOVA | Analysis of variance |
| *Adj.SS* | Adjusted sum of squared deviation |
| *Adj.MS* | Adjusted mean squared deviation |
| *CCD* | Central composite design |
| *DOF* | Degree of freedom |
| *GA* | Genetic algorithm |
| *L/H* | Relative length |
| N–S | Navier–Stokes |
| *Q* | Heater power (W) |
| RSM | Response surface methodology |
| *RSME* | Root mean square errors |
| $R^2$ | Determination coefficients |
| $R^2$(Adjusted) | Adjusted determination coefficients |
| $RH_{max}$ | Maximum relative humidity (%) |
| *T* | Environmental temperature (K) |
| $T_{min}$ | Minimum temperature inside the chamber (K) |
| $T_{heater}$ | Maximum temperature of the heater surface (K) |
| *W/H* | Relative width |
| *y* | Response value |
| *ε* | Prediction error |

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
