# Peer review of "Numerical Study on the Heating Effect of a Spring-Loaded Actuator—Part II: Optimization Design of Heater Parameters"

_actuators, doi:10.3390/act12050212_

Round 1
Reviewer 1 Report
Review Manuscript ID: actuators-2347504
Title: Numerical Study on the Heating Effect of a Spring-Loaded Actuator—Part II: Optimization Design of Heater Parameters
Authors: Zhen Zhao, Lei Xi, Jianmin Gao, Liang Xu and Yunlong Li
The main subject of this document is a study on the temperature and relative humidity calculation and heater optimization design of a typical spring actuator heating system using numerical simulation methods.
The article abstract describes a study on the optimization of the heating system of a spring actuator to ensure its safe operation. However, there are several issues with the review of the article. Firstly, it does not provide any context or background information on the significance of the study, making it difficult to understand the motivation behind the research.
Additionally, the abstract lacks clear objectives and research questions, which are essential components of any research study. The methods used in the study are briefly described, but the details are insufficient, and the results are presented without any discussion or analysis. Furthermore, the abstract does not provide any information on the limitations or implications of the study, making it difficult to assess the significance of the findings. Therefore, the abstract needs to be revised to address these issues and provide a clear understanding of the research study.
Although the work presents concordance between what is described in the abstract and the conclusions, the results explained in Section 4 (line 243) analysis of results and discussion make it difficult to follow up on what has been done.
It should be borne in mind that the present work complements a previous article oriented to the numerical modelling of heat and mass transfer problems. This presents the profile of a work of statistical, mathematical exercise of adjustments of previously obtained values, but not the directives of a real optimization of a problem by another complex as the one treated in the article part I.
The main question is, what is the best design? The manuscript's title uses "optimization" term and should be straightforward to give the best configuration.
Comments are provided below that justify what has been previously expressed.
Major comments
For example, how is a Tmin^0.5 understood? Does it represent the square root of the minimum temperature? What physical sense does this variable have?
What is the justification for the variable RHmax to be transformed into a natural logarithm function?
Moreover, what is the definition of Adj.SS or Adj.MS? Although it is included in the table of nomenclature, the reasons for its use and definition should be explained.
In line 187 the authors say that design methods based on surface responses are widely used for the construction of approximate models. The reason for this statement is not justified, and no references are provided to justify it.
Among the advantages that are given, all of them could be applied to heat transfer models. But the bases that govern them do not include the accumulated heat by volume, they do not take into account radiation. It is also accepted that there is a high probability that a polynomial of the second or third degree adapts to otherwise universal values.
Provide references that justify the universality of the expression (4), line 198.
Regarding the restrictions imposed on Q <=500 (line 231) then in the conclusions section (line 406) there is talk of powers of 910 W. How do you justify heating values higher than the restrictions?
Drafting and sentences aspects
In line 372 "of the" maintains the format of the italicized relation of the quotient of variables.
The online 373 "from This ..." makes no sense and appears to be an incomplete sentence.
Regarding Fig 5, the reasons for its inclusion and how the message of the same should be interpreted within the method that is intended to justify must be expanded.
In line 208 refers to a software that does not give references of the version or scope of the methods that it allows to apply.
Figures 9, 10 and 11 are very difficult to read as the resolution used is insufficient.
Nor is it explained what meaning "Standardized Effect" has and what the terms A, B, AB, AA and the following mean.
Author Response
See documentation for details.

Reviewer 2 Report
In this article, the authors have numerically studied the characteristics of a spring-loaded actuator heating system and have optimized the heater design. I have the following queries the authors need to address before making a decision.
- The authors need to present the model equations and boundary conditions more clearly. What does fb,i represent in Eq. (2). Its mathematical form should be mentioned in the article. Moreover, the present form of the energy equation is incorrect.
- The authors should consider presenting the mesh model in Fig. 3 in a more convenient way. The present version does not make much sense.
- It is unclear how relative humidity is estimated from the model equations.
- The image quality of Figs. 9-11 need improvement. The figure legend is not quite clear in the present version.
- It is not clear what the authors want to convey through the statement “when the environmental temperature decreases from 233K to 313K” in lines 408-409.
The manuscript needs to be thoroughly revised for language usage. Several grammatical inconsistencies are noted in the present version.
Author Response
See documentation for details.

Reviewer 3 Report
Please see attached PDF for comments and suggestions for authors.

Author Response
See documentation for details.

Reviewer 4 Report
The paper provides with numerical modelling and optimisation of thermally controlled spring-loaded actuator. The novelty of presented research may be thought as combining temperature and humidity effects produce by the actuator heating system.
The following remarks should be taken into account in the revised version of the manuscript:
Major remarks:
1. Novelty of proposed research is related to the effect of humidity since temperature effect on the actuator material has been studied in previous numerous papers. Therefore there is a clear need to concisely present the physical aspects of the effect of humidity on the spring loaded actuator operation and especially the actuator material.
2. There should be clarified humidity control system for the discussed testing stand that has been applied for the model validation.
3. In the optimisation procedure the effect of relative humidity has been taken into account along with heater thermal capacity. However, the heater capacity might be potentially easy controlled, but in the case of relative humidity such control is problematic. These aspects should be clarified.
Minor remarks:
1. In my opinion presentation of conventional governing equations, eq. (1) – (3) is not necessary, in addition these equations have not been precisely described. Therefore, I suggest to reduce presentation of these equations and jus t provide appropriate references only.
2. The references numbering should be placed just after the names of the authors of citied papers in order to clarify presentation.
3. Presentation of the model validation: maximum deviation for temperature should not be provided in per cents but in temperature unit.
Author Response
See documentation for details.

Round 2
Reviewer 1 Report
Review Manuscript ID: actuators-2347504
Numerical Study on the Heating Effect of a Spring-Loaded Actuator—Part Ⅱ: Optimization Design of Heater Parameters
Authors: Zhen Zhao, Lei Xi, Jianmin Gao, Liang Xu, Yunlong Li
Dear Authors,
As a reviewer for this manuscript, I was thoroughly impressed with the author's dedication to enhancing the quality of their work. The research is meticulous, and the results are presented clearly and compellingly.
Additionally, the author has made commendable efforts to address the comments and suggestions provided during the review process, resulting in significant improvements to the manuscript.
Considering the overall quality of the work, I would like the acceptance of this manuscript for publication.
Author Response
Thank you very much for the recognition of the reviewers.
Reviewer 2 Report
The authors have satisfactorily addressed the previous comments. However, a few new issues have been noticed in this revised manuscript based on the changes made over the earlier version.
Equation 1 is presented in an incorrect form. The mass fraction is a scalar quantity. However, the div operator takes a vector field as an input and returns a scalar field.
Several typological and grammatical mistakes are noted in this revised version. For instance; line 198, total grid “size” is 6.6 million; line 153, control equations are “discrete” by…. and so on. The authors should thoroughly revise the manuscript for such inconsistencies.
As indicated above, the authors must thoroughly check this revised manuscript for grammatical misrepresentation.
Author Response
See the document for details.

Reviewer 3 Report
The authors have well-addressed the comments and suggestions. The paper is now ready for publication. Congrats; good jobs from the authors!
Author Response

(The authors gave the same response as above.)
